# Human Bocavirus Circulating in Patients with Acute Gastroenteritis in Taiwan, 2018–2022

**DOI:** 10.3390/v16101630

**Published:** 2024-10-18

**Authors:** Shu-Chun Chiu, Ya-Chun Yu, Lun-Hao Hsieh, Yu-Hua Chen, Yu-An Lu, Jen-Hung Chang, Jih-Hui Lin

**Affiliations:** Center for Diagnostics and Vaccine Development, Centers for Disease Control, Taipei 11561, Taiwan; schiu@cdc.gov.tw (S.-C.C.); yachung0617@gmail.com (Y.-C.Y.); luenhau@cdc.gov.tw (L.-H.H.); yuhua1688@cdc.gov.tw (Y.-H.C.); yuan@cdc.gov.tw (Y.-A.L.); 4108043027@cdc.gov.tw (J.-H.C.)

**Keywords:** human bocavirus, acute gastroenteritis, Taiwan

## Abstract

Human bocavirus (HBoV) has been identified as a viral agent with a global presence, especially in young patients with gastrointestinal infections. In this study, we aimed to evaluate the epidemiological patterns of the HBoVs associated with acute gastroenteritis (AGE) in Taiwan. A total of 2994 AGE fecal samples from several diarrhea outbreaks from 2018 to 2022 were analyzed. From the samples, 73 positive samples were detected in three different bocaviruses: 30 (41.1%) were from HBoV1; 37 (50.7%) were from HBoV2; and 6 (8.2%) were from HBoV3, revealing the respective prevalences in AGE of 1%, 1.2%, and 0.2%. HBoV1 and HBoV2 were the two major epidemic agents of HBoVs in Taiwan during this study period and have seasonal distinct patterns with an epidemic peak from October to the following March. Phylogeny reconstruction and evaluation were implemented in Mega X; the results revealed that most HBoV1 strains in Taiwan appeared to be closely related to those strains from other Asian countries. The HBoV2 exhibited substantial genetic diversity and the HBoV3 genes showed discordance of groups.

## 1. Introduction

Human bocaviruses (HBoVs) belong to the *Parvovirinae* subfamily and are classified within the *Bocaparvovirus* genus. The virions of HBoVs are small and lack an envelope. The genome is composed of linear, single-stranded DNA approximately 5.5 kb in length and featuring short terminal hairpins [1]. Furthermore, the genome consists of three open reading frames, encoding nonstructural protein (NS1), noncapsid protein (NP1), and viral capsid proteins (VPs; VP1/VP2/VP3) [2,3]. Although HBoVs are generally considered a conserved virus, the VPs region exhibits diversity, often displaying variations in the amino acid sequences. By contrast, NS1 and NP1 are the most conserved regions [4].

HBoVs are known to be potential agents of acute gastroenteritis (AGE) and can be identified in various bodily fluids, including respiratory samples [5], blood [6], saliva [7], urine [8], and feces [9]. On the basis of genetic diversity, bocaviruses comprise four HBoVs: HBoV1, HBoV2, HBoV3, and HBoV4 [2]. The nomenclature for HBoV1 was established based on its sequence similarities and genomic organization, which closely resemble those of bovine parvovirus and canine minute virus. HBoV1 was initially discovered in children with lower respiratory tract infections in Sweden and was more commonly detected in respiratory specimens than in stool samples [10,11]. Variants with sequences similar to multiple regions of the HBoVs genome were subsequently identified, leading to the designations HBoV2, HBoV3, and HBoV4. These variants were mainly found in stool samples and corresponded to the NS1, NP1, and VPs regions, respectively [5].

HBoVs have been identified as viral agents with a global presence, and numerous studies have reported their potential involvement in causing diarrhea outbreaks [5,12,13,14,15,16,17,18]. HBoV1 primarily infects the respiratory system, leading to symptoms such as fever, coughing, rhinorrhea, wheezing, and gastroenteritis [5,19]. By contrast, HBoV2–HBoV4 are more frequently detected in patients with gastrointestinal problems [2,20,21]. To date, the prevalence of diarrhea cases and the mortality rate among children with diarrhea remain high [22]. In Asia, surveillance studies have been conducted in multiple countries and territories, including Thailand [23], Bangladesh [24], Hong Kong [11], Japan [25], China [26], and South Korea [27]. However, data regarding the prevalence of HBoVs infections in Taiwan are limited. Therefore, this current study evaluated the epidemiological patterns of the aforementioned HBoVs associated with AGE in Taiwan to facilitate the genetic characterization of these strains.

## 2. Materials and Methods

### 2.1. Sample Collection

In this study, an outbreak was defined as an instance involving 2 or more cases of gastroenteritis that were correlated in terms of their location and timing. A new outbreak was defined as one that occurred at least 7 days after the most recent case in a prior outbreak or in a different geographic area, following WHO guidelines [28]. The gastrointestinal symptoms encompassed vomiting, fever, and diarrhea, which was defined as 3 or more instances of loose or watery stools within a 24 h period. Stool samples from episodes of AGE were collected between 2018 and 2022. The biological materials used in this study adhered to standard diagnostic procedures following recommendations provided by medical professionals and without any modification to the established sampling protocol. Following the regulations of the Taiwan Centers for Disease Control (CDC), the procedure did not require specific consent from patients.

### 2.2. HBoVs Detection

Fecal samples obtained from patients were sent to the Taiwan CDC for comprehensive bacterial and viral testing. The bacterial tests encompassed the identification of 7 pathogens, namely *Salmonella*, *Shigella*, *Vibrio cholera*, *Vibrio parahaemolyticus*, pathogenic *Escherichia coli*, *Staphylococcus aureus*, and *Bacillus cactus*. The viral tests focused on the detection of norovirus, rotavirus, sapovirus, and bocavirus. For the extraction of viral DNA from multiple outbreaks, the taco Nucleic Acid Automatic Extraction System (GeneReach, Taichung City, Taiwan) was employed. The extracted DNA was stored at −80 °C until used for analysis. The detection of the 4 HBoVs involved the use of a nested polymerase chain reaction (PCR) approach, targeting the VPs region according to the method outlined by Kapoor et al. [2]. In brief, the first-round PCR primers were AK-VP-F1 (5′-CGC CGT GGC TCC TGC TCT-3′) and AK-VP-R1 (5′-TGT TCG CCA TCA CAA AAG ATG TG-3′), and the second-round PCR primers were AK-VP-F2 (5′-GGC TCC TGC TCT AGG AAA TAA AGA G-3′) and AK-VP-R2 (5′-CCT GCT GTT AGG TCG TTG TTG TAT GT-3′). The PCR products were verified through electrophoresis on a 1.5% agarose gel, with the expected amplicon size being 576 bp.

### 2.3. Sequence Analysis and Phylogenetic Characterization

For sequence analysis, specific regions within the NS1, NP1, and VPs genes were amplified and sequenced. These nucleotide sequences were then compared with those of reference strains available in the NCBI GenBank database by using the BLAST tool. To explore phylogenetic and molecular evolutionary relationships, Molecular Evolutionary Genetics Analysis software (MEGA X) was used [29]. Phylogenetic trees representing partial NS1, NP1, VPs genes were constructed using the maximum likelihood (ML) method. The nucleotide substitution models with the best fit were determined on the basis of the lowest Bayesian Information Criterion (BIC) scores; accordingly, the HKY + G model was selected. The robustness of the ML tree was statistically evaluated through a bootstrap analysis by using 1000 bootstrap samples. All the nucleotide sequences of the HBoV obtained in this study were submitted to the GenBank and have been assigned the accession numbers PP460959-PP460993.

## 3. Results

### 3.1. Demographics and Prevalences of HBoVs

During this study period, namely 2018 to 2022, a total of 2994 patients were analyzed. Among these individuals, 1683 (56.2%) were male, and 1311 (43.8%) were female. The age range of the participants ranged from 1 month to 94 years. The most common clinical symptoms reported were vomiting and diarrhea. Of the 73 samples that were confirmed positive for HBoVs in the laboratory, no cases of dual infections were observed. Among these 73 positive samples, 40 (54.8%) were from male patients, and 33 (45.2%) were from female patients. The partial VP1 region of all 73 HBoV-positive strains identified in this study was successfully amplified and sequenced, and the corresponding types were determined through sequence analysis. Over the course of the 5-year study period, three HBoVs were identified, with the prevalence rates of HBoV1, HBoV2, and HBoV3 being 41.1%, 50.7%, and 8.2% in HBoV-positive samples, respectively (Table 1).

The relative prevalence of HBoVs DNA in the stool samples exhibited yearly variations (1.5% to 4.1%). Notably, HBoV2 emerged as predominant in HBoV-positive samples from 2018 to 2020, with viral detection rates of 80% in 2018, 68% in 2019, and 70% in 2020. By contrast, HBoV1 was predominant from 2021 to 2022, with viral detection rates of 88.9% in 2021 and 78.9% in 2022. Both HBoV1 and HBoV2 were detected in every year of this study period, whereas the HBoV3 was detected only in 2019 and 2020. This pattern indicates that HBoV1 and HBoV2 were the major agents of HBoV-positive samples circulating in patients with acute diarrhea in Taiwan during this study period, with a peak in prevalence observed from October to the following March each year (Figure 1).

Throughout this entire study period, both HBoV1 and HBoV2 were consistently detected each year. By contrast, the HBoV3 genotype was identified only in 2019 and 2020. Notably, although the overall prevalence of the HBoV2 infection was higher than that of HBoV1 (50.7% versus 41.1%, respectively), an analysis of the annual prevalences from 2018 to 2022 revealed a noteworthy pattern: HBoV1 and HBoV2 alternated as the most predominant types of HBoV-positive samples throughout 2021 and 2022, with prevalences of 88.9% and 78.9%, respectively.

### 3.2. Age Distribution of HBoVs

The age range of the 73 HBoV-positive patients ranged from 5 months to 60 years. Notably, a significant proportion of these cases, specifically 87.7% (64 of 73), were detected in children under 11 years of age. When these 73 patients were divided into four age groups, namely ≤3 years, 4–11 years, 12–17 years, and ≥18 years, the highest infection rate was observed in the group of individuals under 3 years old, with a prevalence of 46.6%, followed by the 4–11 years group, with a prevalence of 41.1%. Among those aged 12–17 years, 8.2% (6 of 73) were detected as positive for HBoVs, and among those aged ≥18 years, 4.1% (3 of 73) tested positive for HBoVs.

We analyzed the distribution of the HBoVs across the aforementioned age groups. The results revealed that HBoV2 was detected in all the age groups, with infection rates ranging from 41.2% to 100%. HBoV1 was identified in all the age groups except for ≥18 years, with infection rates ranging from 33.3% to 50%. By contrast, HBoV3 was detected only in the ≤3 years and 4–11 years groups, with considerably lower infection rates ranging from 9% to 10% (Table 2). The most common clinical manifestations were vomiting and diarrhea; fever and nausea were less frequently observed.

### 3.3. Seasonality of HBoVs

The relative prevalence of HBoVs varied across the seasons, and the monthly distribution of HBoVs infections from 2018 to 2022 is presented in Figure 1. Notably, the monthly analysis of the HBoVs detection throughout the 5-year study period indicated that HBoV occurred annually in Taiwan, with a peak from October to March. Furthermore, the different HBoVs were predominant in different years; HBoV2 dominated in 2018 to 2020, whereas HBoV1 was the most prevalent in 2021 and 2022. HBoV3 was relatively rare throughout this study period, with only six HBoV3-positive cases, all detected in October and November.

### 3.4. Phylogenetic Analysis of HBoVs

The initial phylogenetic trees were constructed using the ML method implemented in MEGA X. The results, based on the VPs gene of the HBoVs detected in Taiwan (i.e., HBoV1, HBoV2, and HBoV3), revealed that most of the HBoV1 strains appeared to be closely related to strains from Asian countries, including Japan, China, and Vietnam (Figure 2). By contrast, all 17 of the HBoV2 strains in Taiwan were categorized as HBoV2A, with none categorized as HBoV2B. Furthermore, HBoV2 in Taiwan exhibited substantial genetic diversity, with nucleotide sequence identities ranging from 95.7% to 99.9%. Furthermore, the six observed strains of HBoV3 were divided into two clades: one closely related to the HBoV3 prototype strain reported in Australia in 2001 (NC_012564/HBoV3/Australia/2001), and the other consisting of three HBoV3 strains from Taiwan.

## 4. Discussion

HBoVs was first reported in Taiwan in 2009 in patients with respiratory tract infections [30]. Several studies have indicated the presence of genetically distinct HBoVs in fecal samples from patients with AGE [20,24,31,32]. In the present study, the prevalence of a HBoVs infection in patients with AGE was investigated in Taiwan from 2018 to 2022. Multiple HBoVs were identified by detecting the VPs region. Over the course of the 5-year study period, HBoV2 is the most common with incidences up to 50.7%, followed by HBoV1 (41.1%) and then HBoV3 (8.2%). Notably, HBoV4 was not detected in any of the analyzed patients during the study period. Both HBoV1 and HBoV2 were identified in every year, with HBoV2 being the most common of HBoV-positive cases in 2018 (80%), 2019 (68%), and 2020 (70%) and with HBoV1 being the most common in 2021 (88.9%) and 2022 (78.9%). HBoV3 was observed in only four patients in 2019 and in two patients in 2020. Notably, a predominant type of HBoV was detected in each year of this study period, with HBoV2 being the most common circulating type in 2018–2020 and with HBoV1 being the predominant circulating type in 2021–2022 (Table 1). Additionally, our data suggested an almost equal prevalence of HBoV1 and HBoV2 (40.1% versus 50.7%, respectively) in HBoV-positive samples and a much lower prevalence of HBoV3. These findings indicated that HBoV1 and HBoV2 were the major HBoV types in Taiwan during the study period.

A genetic analysis of the HBoVs genes revealed distinct phylogenetic relationships among the NS1, NP1, and VPs genes of HBoV1, HBoV2, and HBoV3. Specifically, the VPs gene of HBoV1 formed a separate clade that was distinct from HBoV2 and HBoV3, and the NS1 and NP1 genes of HBoV1 and HBoV3 clustered closely together but differed from those of HBoV2 (Figure 2). Phylogenetic analysis further indicated that the most prominent sequence variations occurred in the VPs gene. Furthermore, our results suggest the presence of several monophyletic groups with relatively long branch lengths that consisted exclusively of Taiwanese HBoV2 isolates; this observation implies that this particular virus may have been circulating in Taiwan for an extended period.

The seasonality of HBoVs infections in patients with acute diarrhea in Taiwan exhibits a distinct pattern, namely a peak in October to the following March. Such seasonality coincides with the norovirus epidemic season and contradicts the findings of previous reports that have suggested that HBoVs infections occur all year round [11,23]. In our study, HBoV1 was detected more frequently in wintertime in 2021–2022, similar to previous studies showing that the seasonality of HBoV1 in NPAs and fecal samples was similar [11,33], while HBoV2 was detected all year round in 2018–2020, showing no significant seasonal pattern, similar to previous studies [34,35]. Although the primers used in the present study were capable of detecting HBoV4, no HBoV4 cases were detected. HBoV2 and HBoV1 exhibited nearly equally high prevalences, underscoring the importance of monitoring both HBoV1 and HBoV2 over other HBoVs. The present findings support the hypothesis that HBoVs infections are relatively frequent in children with AGE. However, further studies are required to determine whether HBoV plays a causative role in AGE or whether it exacerbates the severity of infections caused by other pathogens.

## Figures and Tables

**Figure 1 viruses-16-01630-f001:**
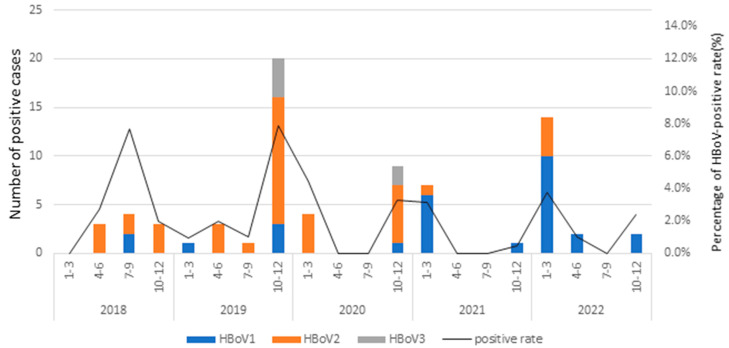
Seasonal distribution of HBoV-positive rates in Taiwan, 2018–2022.

**Figure 2 viruses-16-01630-f002:**
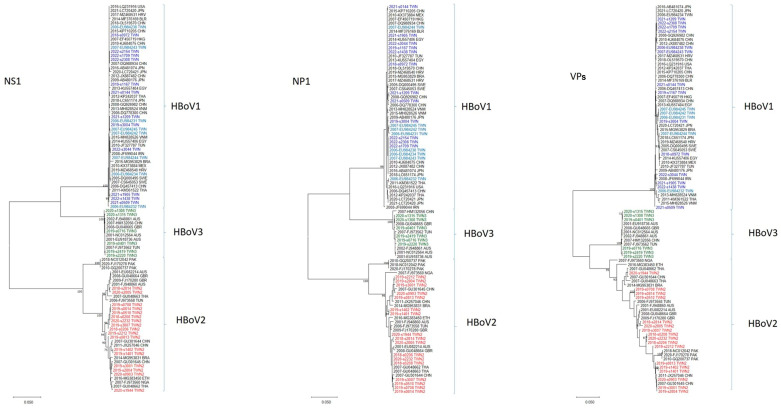
Phylogenetic analysis of full-length HBoV-positive patients NS1, NP1, VPs genes using the maximum likelihood method. Stability of tree topology was evaluated by using 1000 bootstrap replicates. Only bootstrap values greater than 75% are shown on the branch.

**Table 1 viruses-16-01630-t001:** The number of HBoV-positive samples in diarrhea patients in Taiwan, 2018–2022.

Year of Study	Number of Samples	Gender		Number of HBoV-Positive	Gender		Species		
		Male	Female		Male	Female	HBoV1	HBoV2	HBoV3
2018	507	303	204	10 (2.0%)	6 (2.0%)	4 (2%)	2 (20%)	8 (80%)	0
2019	607	355	252	25 (4.1%)	14 (3.9%)	11 (4.4%)	4 (16%)	17 (68%)	4 (16%)
2020	569	319	250	10 (1.8%)	3 (0.9%)	7 (2.8%)	1 (10%)	7 (70%)	2 (20%)
2021	595	312	283	9 (1.5%)	5 (1.6%)	4 (1.4%)	8 (88.9%)	1 (10.1%)	0
2022	716	394	322	19 (2.7%)	12 (3%)	7 (2.2%)	15 (78.9%)	4 (21%)	0
Total	2994	1683	1311	73 (2.4%)	40 (2.3%)	33 (2.5%)	30 (41.1%)	37 (50.7%)	6 (8.2%)

**Table 2 viruses-16-01630-t002:** Age distribution and symptoms of HBoV-positive cases with AGE in Taiwan, 2018–2022.

Age Groups	Number of HBoV-Positive		Species		Symptoms					
		HBoV1	HBoV2	HBoV3	Diarrhea	Abdominal Pain	Fever	Vomiting	Nausea	Dizziness
≤3	34 (46.6%)	17 (50.0%)	14 (41.2%)	3 (9%)	19	8	4	20	3	0
4–11	30 (41.1%)	10 (33.3%)	17 (56.7%)	3 (10%)	15	12	5	20	4	0
12–17	6 (8.2%)	3 (50%)	3 (50%)	0	2	4	0	2	0	0
>18	3 (4.1%)	0	3 (100%)	0	2	2	1	1	1	1
Total	73	30	37	6	38	26	10	43	8	1

## Data Availability

The data presented in this study are all presented in the manuscript.

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
