# Peer review of "Human Bocavirus Circulating in Patients with Acute Gastroenteritis in Taiwan, 2018–2022"

_viruses, 2024, doi:10.3390/v16101630_

Round 1
Reviewer 1 Report
Comments and Suggestions for Authors
In this manuscript, the authors performed an epidemiological study on human Bocavirus (HBoV) in patients with acute gastroenteritis in Taiwan between 2018 and 2022. The researchers used phylogenetic analysis to identify the genotypes of HBoV and investigated their prevalence over the 5-year period. The study highlights the seasonal distribution of the three HBoV genotypes identified, as well as the age distributions. Interestingly, the authors found a seasonality pattern of HBoV, where higher incidence was observed between October and March, which contradicted with current literature. This is a short paper that provide an insight into the circulating HBoV infections in Taiwan.
In the introduction, HBoV1 was stated as more commonly detected in respiratory specimens compared to stool samples. However, in this study, which analysed stool samples, HBoV1 was found to be the second most prevalent genotype. Could the authors elaborate on this in the discussion? Furthermore, there is little to no discussion, the authors simply summarised their results but lacked the discussion of results with existing literature.
E.g. what could be the reason for this emergence of HBoV1 from 2021 onwards?
Is this a common phenomenon for one genotype to dominate for a few years before the dominance of another? Similar to e.g. norovirus?
Despite the low presence, why is HBoV3 only observed in Oct-Dec, as compared to other genotypes?
I would suggest the authors to discuss their results in relation to the current literature.
Minor comments:
Line 52: I would suggest replacing the word “Accordingly” to “Therefore”.
Line 59: why was the outbreak arbitrarily defined? The authors should follow what is currently used to measure as a “new” outbreak in the literature to make it comparable to other studies.
Line 60: commas should be used instead of semicolons.
Author Response
Comments 1: In the introduction, HBoV1 was stated as more commonly detected in respiratory specimens compared to stool samples. However, in this study, which analysed stool samples, HBoV1 was found to be the second most prevalent genotype. Could the authors elaborate on this in the discussion? Furthermore, there is little to no discussion, the authors simply summarised their results but lacked the discussion of results with existing literature.
Response 1: Thank you for pointing this out. We agree with this comment. Therefore, we have added more discussion in our literature. This changes can be found in line 213-221.
Minor comments:
Comments: Line 52: I would suggest replacing the word “Accordingly” to “Therefore”.
Response : We agree reviewer’s suggestion and have changed.
Comments: Line 59: why was the outbreak arbitrarily defined? The authors should follow what is currently used to measure as a “new” outbreak in the literature to make it comparable to other studies.
Response: We agree reviewer’s comments and re write the sentence in the text (line60-62).
Comments: Line60: commas should be used instead of semicolons.
Response: corrected.
Reviewer 2 Report
Comments and Suggestions for Authors
This study aimed to evaluate epidemiological patterns of the four human bocaviruses, HBoV1-4, in patients (1 month to 94 years of age) with acute gastroenteritis (AGE) in Taiwan during a five-year period, 2018-2022. Out of 2994 stools examined, 73 (2.4%) exhibited HBoV DNA by a VP-nested PCR, developed by a reputable group in the US. By sequencing of both the non-structural and structural genes, they found varying yearly proportions of the different HBoVs, with clear predominance of HBoV1 and 2, a few HBoV3, but none of the rare HBoV4; in total HBoVs varying from 1.5% to 4.1% during the five years, with main peaks during October to March. The most affected age group was the smallest children <3 years of age. This study gives valuable information on the occurrence of the four different bocaviruses in AGE. However, there were many erroneous statements that need to be corrected, and the individual HBoV1-4 prevalence rates in AGE need to be presented, as commented below.
General concerns:
1. The viral terminology is flawed, especially “genotypes” and “species” should be corrected throughout the manuscript text and tables:
HBoV1-4 are neither species nor different genotypes of a HBoV species! According to ICTV, HBoV1 and HBoV3 belong to the same species Primate bocaparvovirus 1 (or along the new binomial name, Bocaparvovirus primate1), whereas HBoV2 and HBoV4 both belong to another species, Primate bocaparvovirus 2 (or Bocaparvovirus primate2), taxonomy following the NS-1 amino acid sequences. (such taxon species names should also be written in italics, like the genus and family taxons, which is not visible in this website)
HBoV1-4 should thus rather be called just “bocaviruses”, or “HBoVs”.
- On the other hand, HBoV1 and HBoV3 could be called genotypes of their common species Bocaparvovirus primate1 and HBoV2 and HBoV4 as genotypes of their common species Bocaparvovirus primate2, but this is too complicated.
- Ref 2, Kapoor et al., correctly discussed “HBoV2 genotypes”, meaning HBoV2A and B, not “HBoV genotypes”, which is a different thing. (Erroneously though, they referred to HBoV1-4 as species, which they are not)
2. Also, the concept of “prevalence” or “detection rates” should be corrected. In many places the Authors gave extremely high prevalences or detection rates for HBoV1 and HBoV2, of up to 88.9%. However, these are not true prevalence rates in AGE, but only proportions of HBoV1 or HBoV2 among the 73 total HBoV positives. This gives the reader a very wrong picture of the true prevalence rates of HBoV1 and HBoV2 DNA in AGE, which in fact were only around 1% for both. Nowhere is this true prevalence mentioned! This should be clarified and corrected, starting from the Abstract and including the Tables! See also specific comments 9 and 11 below.
3. The Discussion is merely a repetition of the results. More comparisons with other similar studies regarding prevalence, age, seasonality etc. should be included and repetitions of results deleted.
4. The references are not always up-to-date concerning basic facts and more other similar studies need to be cited in the Discussion. There is also a newer ICTV paper of parvovirus taxonomy.
5. The conclusions are misleading, see comment 13 below.
Specific comments:
Errors in Abstract:
1. “Human Bocavirus” should be written with a small letter “b”: “bocavirus”.
2. Please correct lines 13-14: “Of them, 73 positive samples…”. This sentence contains four errors. i) This would imply that of the 2994 samples only 73 were of HBoVs 1, 2 or 3, but others could be of e.g., HBoV4. ii) “samples were detected in 3 different HBoV genotypes” does not mean anything, you cannot detect “samples in genotypes”? iii) you cannot say “genotypes were from HBoV1”. iv) delete the word “genotypes”, as explained above, HBoV1-4 are not genotypes. The sentence could thus be corrected to: “Among them, 73 samples were positive for HBoV DNA, of which 30 (41.1%) were of HBoV1, 37 (50.7%) of HBoV2, and 6 (8.2%) of HBoV3”. It would further be correct to tell the true prevalences here too by continuing this sentence with: “, revealing the respective prevalences in AGE of 1%, 1.2% and 0.2%.” (see General comment 2 above and Specific comments 9 and 12 below)
Errors in Introduction (and throughout the manuscript):
3. HBoVs have actually three structural proteins, not two as stated: VP1, VP2 and VP3, in the order of length, due to the detection of a middle truncated protein, VP2, so the former VP2 is actually now referred to as VP3, which is the major protein forming the capsid. Please cite a more recent publication or review, where this is clarified.
4. “HBoV is known to be a causative agent of acute gastroenteritis” (line 31). This is not true, it is still much debated, many publications show the contrary, so this statement should be re-written. The Authors actually later state (line 44): “potential involvement in causing diarrhea outbreaks” which is much better. However, they next state (line 47): “primarily associated with gastrointestinal problems”, which again is not true, HBoVs are detected also in stools from individuals without AGE, so there does not seem to be any true “association” with AGE.
5. Furthermore, “HBoV” (line 31 and elsewhere) is not one virus, but four, of which one is a respiratory virus, HBoV1, and two others, HBoV2 and 3, are enteric, whereas HBoV4 is too rare to know its main site of infection, but has been found in a few stools. Throughout the manuscript, it should be specified which HBoV is meant where, and if all HBoVs are meant, it should be written as “HBoVs” in plural, since they are different viruses.
6. HBoV is not “divided into 4 genotypes” (line 33). All four are not “variants of HBoV” either, because “HBoV” is not a single species. They are different viruses, belonging to 2 species, as mentioned in the above General comments.
Errors in Results (and elsewhere):
7. “Of 73 samples that were confirmed positive” (line 104), This is also ambiguous, as in the Abstract, but could here be overcome by inserting “the”: “Of the 73 samples that were confirmed positive”.
8. Lines 107-109: “The partial VP1 region of all 73 HBoV-positive strains identified in this study was successfully amplified and sequenced, and the corresponding genotypes were determined through sequence analysis.” Perhaps it would be better to claim that this differentiation of the four HBoVs would be done based on also the NS1 gene (as shown in Fig. 2), since the taxonomy relies on NS1.
However, on lines 87-88, “For sequence analysis, specific regions within the NS1, NP1, and VP1/VP2 genes were amplified and sequenced”, it is not stated which and how long “specific regions” were sequenced and analysed in each gene in the Fig. 2 phylogenetic tree. The figure legend implies that they were full-length genes? Please clarify both in text and figure legend.
9. Lines 109-110. “Over the course of the 5-year study period, 3 genotypes of HBoV were identified, with the prevalence rates of HBoV1, HBoV2, and HBoV3 being 41.1%, 50.7%, and 8.2%, respectively”. This is totally incorrect, the true prevalence rates of HBoV1, 2 and 3 DNA in the 2994 stool samples were 1%, 1.2%, and 0.2% (calculated by myself since I could not find these true prevalences anywhere). Those given numbers and percentages are the proportions of HBoV 1, 2 and 3 among the 73 HBoV-DNA positives. The same flaw is also on lines 116-117: “with viral detection rates of 80% in 2018, 68% in 2019, and 70% in 2020” and “with viral detection rates of 88.9% in 2021 and 78.9% in 2022”, and on lines 125-129: “the overall prevalence of HBoV2 infection was higher than that of HBoV1 (50.7% versus 41.1%)”, and on lines 140-146: “The results revealed that HBoV2 was detected in all the age groups, with infection rates ranging from 41.2% to 100%. HBoV1 was identified in all the age groups except for ≥18 years, with infection rates ranging from 33.3% to 50%”. Finally, all these repeated in Discussion (lines 182-192)!? These are not true! Please re-write for clarity and include the correct prevalence and detection rates among the 2994 AGE patients! See also comment 11 below and General comment 2 above.
10. Line 114: “The relative prevalence of HBoV exhibited seasonal variations (1.5% to 4.1%)”. This should be “yearly variations”. “Seasonal” means the months or the four seasons within a year. Moreover, more exactly it should state: “The relative prevalence of HBoV1-4 DNA in stool samples exhibited yearly variations” (or “of total HBoV DNA” if this means all four viruses)
11. Tables 1 and 2: Exchange “Species” with “Viruses”. Three misspellings in “HBoV-posiive” in the Table 2 heading, it should be “HBoV positives” or “HBoV-positive stools”, also in Table 1. In Table 2, provide a new column with the number of samples tested per each age group (as done in Table 1 for each year), and another column with the correct prevalence rates within these total numbers of samples for each bocavirus in both Tables 1 and 2, not only the rates within the 73 HBoV positives. See also Specific comment 9 and General comment 2 above. The column headings in Table 2 should be on the same page as the rest of the table.
12. Discussion, line 195: What is mean by: “Each individual HBoV gene was constructed”. Please clarify or re-write.
13. Conclusions, line 212: “The present findings support the hypothesis that HBoV may play a role in the pathogenesis of AGE”. This is misleading, this study did not include non-AGE controls, so nothing can be said of etiology, especially with the low 0.2 to 1% prevalence rates in AGE.
Comments on the Quality of English Language
The English language could be improved at places, some of which have been pointed out in the comments, the remaining are minor issues. In general the language is well understood.
Author Response
Comments 1: The viral terminology is flawed, especially “genotypes” and “species” should be corrected throughout the manuscript text and tables:
Response 1: Thank you for pointing this out. We agree with this comment. Therefore, we have corrected the description from genotypes, species to bocaviruses and HBoVs in the literature.
Comments 2: Also, the concept of “prevalence” or “detection rates” should be corrected. In many places the Authors gave extremely high prevalences or detection rates for HBoV1 and HBoV2, of up to 88.9%. However, these are not true prevalence rates in AGE, but only proportions of HBoV1 or HBoV2 among the 73 total HBoV positives. This gives the reader a very wrong picture of the true prevalence rates of HBoV1 and HBoV2 DNA in AGE, which in fact were only around 1% for both. Nowhere is this true prevalence mentioned! This should be clarified and corrected, starting from the Abstract and including the Tables! See also specific comments 9 and 11 below.
Response 2: We agree reviewer’s opinions. Therefore, we have corrected “prevalence” in the text and described as “the prevalence of HBoV-positive samples” in AGE.
Comments 3: The Discussion is merely a repetition of the results. More comparisons with other similar studies regarding prevalence, age, seasonality etc. should be included and repetitions of results deleted.
Response 3: Thanks for the comments, we added more discuss to compare with other similar studies. This changes can be found in line 210-216.
Comment 4: The references are not always up-to-date concerning basic facts and more other similar studies need to be cited in the Discussion. There is also a newer ICTV paper of parvovirus taxonomy.
Response 4: References were updated.
Comment 5: The conclusions are misleading, see comment 13 below.
Response 5: response in comment 13.
Specific comments:
Comment 1: Human Bocavirus” should be written with a small letter “b”: “bocavirus”
Response 1: corrected
Comments 2: Please correct lines 13-14: “Of them, 73 positive samples…”. This sentence contains four errors. i) This would imply that of the 2994 samples only 73 were of HBoVs 1, 2 or 3, but others could be of e.g., HBoV4. ii) “samples were detected in 3 different HBoV genotypes” does not mean anything, you cannot detect “samples in genotypes”? iii) you cannot say “genotypes were from HBoV1”. iv) delete the word “genotypes”, as explained above, HBoV1-4 are not genotypes. The sentence could thus be corrected to: “Among them, 73 samples were positive for HBoV DNA, of which 30 (41.1%) were of HBoV1, 37 (50.7%) of HBoV2, and 6 (8.2%) of HBoV3”. It would further be correct to tell the true prevalences here too by continuing this sentence with: “, revealing the respective prevalences in AGE of 1%, 1.2% and 0.2%.” (see General comment 2 above and Specific comments 9 and 12 below)
Response 2 : Thanks for opinion. We agree the reviewer’s comment and we correct and re write the sentences for the prevalence data at line 13-14.
Comment 3: HBoVs have actually three structural proteins, not two as stated: VP1, VP2 and VP3, in the order of length, due to the detection of a middle truncated protein, VP2, so the former VP2 is actually now referred to as VP3, which is the major protein forming the capsid. Please cite a more recent publication or review, where this is clarified.
Response 3: Thank you for the comments, we correct the description and update the reference.
Comment 4: “HBoV is known to be a causative agent of acute gastroenteritis” (line 31). This is not true, it is still much debated, many publications show the contrary, so this statement should be re-written. The Authors actually later state (line 44): “potential involvement in causing diarrhea outbreaks” which is much better. However, they next state (line 47): “primarily associated with gastrointestinal problems”, which again is not true, HBoVs are detected also in stools from individuals without AGE, so there does not seem to be any true “association” with AGE.
Response 4: Thanks for the comment and we re-write the sentences of line 31 and line 47.
Comment 5:Furthermore, “HBoV” (line 31 and elsewhere) is not one virus, but four, of which one is a respiratory virus, HBoV1, and two others, HBoV2 and 3, are enteric, whereas HBoV4 is too rare to know its main site of infection, but has been found in a few stools. Throughout the manuscript, it should be specified which HBoV is meant where, and if all HBoVs are meant, it should be written as “HBoVs” in plural, since they are different viruses.
Response 5: Thanks for the comments, we corrected the description from HBoV to HBoVs in the literature.
Comment 6:HBoV is not “divided into 4 genotypes” (line 33). All four are not “variants of HBoV” either, because “HBoV” is not a single species. They are different viruses, belonging to 2 species, as mentioned in the above General comments.
Response 6: response in General response 1, corrected.
Comment 7: “Of 73 samples that were confirmed positive” (line 104), This is also ambiguous, as in the Abstract, but could here be overcome by inserting “the”: “Of the 73 samples that were confirmed positive”.
Response 7: Thanks for the comments, we corrected the sentence as “Of the 73 samples that…”
Comment 8: Lines 107-109: “The partial VP1 region of all 73 HBoV-positive strains identified in this study was successfully amplified and sequenced, and the corresponding genotypes were determined through sequence analysis.” Perhaps it would be better to claim that this differentiation of the four HBoVs would be done based on also the NS1 gene (as shown in Fig. 2), since the taxonomy relies on NS1.
However, on lines 87-88, “For sequence analysis, specific regions within the NS1, NP1, and VP1/VP2 genes were amplified and sequenced”, it is not stated which and how long “specific regions” were sequenced and analysed in each gene in the Fig. 2 phylogenetic tree. The figure legend implies that they were full-length genes? Please clarify both in text and figure legend.
Response 8: Thank for the opinion. We amplified partial VP1 region fragment to determine the type of HBoVs (described in material and methods section, line 81-86), and used full-length fragment of each genes to perform phylogenetic analysis (correct in figure legend line 178).
Comment 9: Lines 109-110. “Over the course of the 5-year study period, 3 genotypes of HBoV were identified, with the prevalence rates of HBoV1, HBoV2, and HBoV3 being 41.1%, 50.7%, and 8.2%, respectively”. This is totally incorrect, the true prevalence rates of HBoV1, 2 and 3 DNA in the 2994 stool samples were 1%, 1.2%, and 0.2% (calculated by myself since I could not find these true prevalences anywhere). Those given numbers and percentages are the proportions of HBoV 1, 2 and 3 among the 73 HBoV-DNA positives. The same flaw is also on lines 116-117: “with viral detection rates of 80% in 2018, 68% in 2019, and 70% in 2020” and “with viral detection rates of 88.9% in 2021 and 78.9% in 2022”, and on lines 125-129: “the overall prevalence of HBoV2 infection was higher than that of HBoV1 (50.7% versus 41.1%)”, and on lines 140-146: “The results revealed that HBoV2 was detected in all the age groups, with infection rates ranging from 41.2% to 100%. HBoV1 was identified in all the age groups except for ≥18 years, with infection rates ranging from 33.3% to 50%”. Finally, all these repeated in Discussion (lines 182-192)!? These are not true! Please re-write for clarity and include the correct prevalence and detection rates among the 2994 AGE patients! See also comment 11 below and General comment 2 above.
Response 9: response in general comment 2. We correct the prevalence in HBoV-positive samples in the literature.
Comment 10: Line 114: “The relative prevalence of HBoV exhibited seasonal variations (1.5% to 4.1%)”. This should be “yearly variations”. “Seasonal” means the months or the four seasons within a year. Moreover, more exactly it should state: “The relative prevalence of HBoV1-4 DNA in stool samples exhibited yearly variations” (or “of total HBoV DNA” if this means all four viruses)
Response 10: Thank s for opinion, we corrected the sentence as “yearly variations”.
Comment 11: Tables 1 and 2: Exchange “Species” with “Viruses”. Three misspellings in “HBoV-posiive” in the Table 2 heading, it should be “HBoV positives” or “HBoV-positive stools”, also in Table 1. In Table 2, provide a new column with the number of samples tested per each age group (as done in Table 1 for each year), and another column with the correct prevalence rates within these total numbers of samples for each bocavirus in both Tables 1 and 2, not only the rates within the 73 HBoV positives. See also Specific comment 9 and General comment 2 above. The column headings in Table 2 should be on the same page as the rest of the table.
Response 11: We rewrite the sentence and correct the misspellings, and re-write the heading of Table 2. Response in General comment 2.
Comment 12: Discussion, line 195: What is mean by: “Each individual HBoV gene was constructed”. Please clarify or re-write.
Response 12: We rewrite the sentence as “Genetic analysis of HBoV genes revealed distinct phylogenetic among the NS1, NP1...”(line 201)
Comment 13: Conclusions, line 212: “The present findings support the hypothesis that HBoV may play a role in the pathogenesis of AGE”. This is misleading, this study did not include non-AGE controls, so nothing can be said of etiology, especially with the low 0.2 to 1% prevalence rates in AGE.
Response 13: Thanks for the reviewer’s opinion, we rewrote the sentence mentioned that HBoV infection is relatively frequent in children with AGE. (line 218).